# Anxiety Levels among Healthcare Workers during the COVID-19 Pandemic and Attitudes towards COVID-19 Vaccines

**DOI:** 10.3390/vaccines12040366

**Published:** 2024-03-28

**Authors:** Anna Lewandowska, Tomasz Lewandowski, Grzegorz Rudzki, Michał Próchnicki, Aleksandra Stryjkowska-Góra, Barbara Laskowska, Paulina Wilk, Barbara Skóra, Sławomir Rudzki

**Affiliations:** 1Faculty of Medical and Health Sciences, State Vocational University in Tarnobrzeg, Henryk Sienkiewicz Street 50, 39-400 Tarnobrzeg, Poland; 2Faculty of Technical Engineering, State University of Applied Sciences in Jarosław, Czarniecki Street 16, 37-500 Jarosław, Poland; tom_lew@interia.pl; 3Department of Endocrinology, Diabetology, and Metabolic Diseases, Medical University of Lublin, Jaczewski Street 8, 20-090 Lublin, Poland; grzegrudzki@gmail.com; 4I Department of Psychiatry, Psychotherapy and Early Intervention, Medical University of Lublin, Głuska Street 1, 20-439 Lublin, Poland; michal.prochnicki@umlub.pl; 5Department of Oncology, Radiotherapy and Translational Medicine, University of Rzeszow, Rejtan Street 16c, 35-959 Rzeszow, Poland; ostryj@o2.pl; 6Faculty of Healthcare, State University of Applied Sciences in Jarosław, Czarniecki Street 16, 37-500 Jarosław, Poland; barbara.laskowska917@gmail.com (B.L.); welca@poczta.fm (P.W.); slawomir.rudzki@umlub.pl (S.R.); 7Collegium Masoviense, University of Health Sciences in Żyrardów, Narutowicz Street 35, 96-300 Żyrardów, Poland; sumoo@o2.pl

**Keywords:** COVID-19 pandemic, healthcare workers, COVID-19 vaccinations, anxiety level

## Abstract

**Background**: The pandemic has proven to be a particular challenge for healthcare workers, not only in the professional but also individual sense. The COVID-19 pandemic negatively influenced their well-being and caused psychological distress. Undoubtedly, direct contact with sick patients, the fight against the pandemic, and observing the epidemiological situation influenced the attitudes of this group towards COVID-19 and vaccinations. The aim of the study was to analyse the level of anxiety among healthcare workers during the COVID-19 pandemic and to assess attitudes towards vaccinations against COVID-19. **Methods**: The cross-sectional study followed the recommendations of STROBE (Strengthening the Reporting of Observational Studies in Epidemiology). A convenience purposive sampling method was used and the study was led among nurses and doctors employed in healthcare facilities. The study used a survey and the Trait Anxiety Scale SL-C. **Results**: The study included 385 participants, with an average age of 48.41 ± 6.76 years. The nurses constituted 55% of the study group and the doctors 45%. A total of 70% of healthcare workers had over 10 years of work experience. Over half of the subjects (57%) became infected with COVID-19. A total of 85% of respondents have received vaccination. A total of 71% of respondents believe vaccinations are harmless. Frequently, the participants assessed their level of anxiety as moderate. **Conclusions**: Almost all surveyed doctors chose to be vaccinated, while the percentage of vaccinated nurses was significantly lower. As a result, it is possible to conclude that the employment position has a significant influence on the decision to get vaccinated against COVID-19. In self-assessment during the COVID-19 pandemic, most healthcare professionals experienced a moderate level of anxiety. Receiving the COVID-19 vaccination reduced the level of anxiety.

## 1. Introduction

Infectious diseases, including SARS-CoV-2, have been and continue to be a major social problem, but more importantly, a health problem—both physically and mentally. Even if we do not become infected, we are nevertheless exposed to the psychological effects of the COVID-19 pandemic. The emergence of severe infectious diseases causes increased anxiety and fear throughout society, as seen by earlier MERS, H1N1, and SARS epidemics. The increased number of infections and deaths caused by infections have instilled a shared fear of death, infected people, and the disease itself. These circumstances have contributed to the elevated levels of anxiety and fear [1].

The psychological impact of the COVID-19 pandemic on individual, group, and societal health is a multidimensional topic. It is the focus of current in-depth research efforts. The psychological impact of the COVID-19 epidemic on the health of individuals, groups, and society is complicated and multifaceted. It is the focus of ongoing in-depth scientific investigations. What is essential is that, given the pandemic’s potential long-term psychological impact, a comprehensive assessment of the event will not be feasible until many years have passed. However, some authors are already formulating terms that refer to psychological problems related to COVID-19, such as pandemic stressors, COVID-19-related psychological stressors, or post-COVID-19 stress disorder [2].

Without a doubt, the COVID-19 pandemic had a negative impact on the well-being of individuals, groups, and society. Despite variations among particular diagnostic units and studied populations, available epidemiological evidence indicates that around 30% of people experienced mental disorders during the pandemic, and over 50% reported psychological distress [2]. During the pandemic, individuals universally experienced a range of difficult emotions, such as anxiety, danger, frustration, or anger. All these emotions lead to a decline in well-being and life satisfaction, potentially negatively impacting the quality of life and leading to mental health issues. Research shows that anxiety, which is the most common negative emotion, can significantly limit social and cognitive functioning, and increase clinical symptoms of social anxiety, phobia, and depression. The main source of anxiety during the pandemic was the disease and its consequences [3,4,5]. Moreover, the pandemic situation fostered social tensions and conflicts and increased the risk of individually harmful behaviours such as substance abuse, self-harm, suicide, aggression, and violence. Dr David Murphy points out that anxiety and danger are the key issues that society has to face during the pandemic [6]. The significant aspect is the fact that society had to deal with the psychological effects of the pandemic much longer than with the pandemic itself. The fear associated with contracting COVID-19 passes, but the fear caused by pandemic stressors remains [2,3,4,5,7].

Some social groups were more exposed to the psychosocial effects of the pandemic than others, for example, people who contracted the disease, people with chronic somatic diseases, as well as young people who are particularly vulnerable to destabilization, isolation, and the need for lifestyle changes. The latest research indicates that anxiety or sadness experienced by young people was the result of consequences and imposed restrictions rather than the illness itself, while in older adults, negative emotions were caused by health-related worries. During the pandemic, healthcare personnel faced heightened stress and chronic anxiety. They put not just their own lives and health at risk while carrying out their duty, but also the lives of others they care about. Healthcare personnel face daily challenges such as concern of infecting their families, a lack of adequate personal protective equipment, and work stress [2,3,5,7,8,9,10,11,12,13].

Considerations to date reveal how devastating the psychological implications of the pandemic are for the social, economic, and healthcare systems. This obviously implies the need for the introduction of preventive and therapeutic measures at the individual and institutional levels, as well as at the level of government agencies—now and in the future [2]. There are numerous recommendations for implementing possible activities at each level stated, but vaccination is the common action for all of them. Vaccines serve a highly vital role in health, particularly in protecting against infectious diseases such as COVID-19. Despite the fact that vaccinations are proven to be successful, there are still differing perspectives and theories about this approach. Opinions differ not only among the general public but even among doctors [2]. The pandemic has proven to be a particular challenge for healthcare workers, not only in the professional but also individual sense. Undoubtedly, direct contact with sick patients, the fight against the pandemic, and observing the epidemiological situation influenced the attitudes of this group towards COVID-19 and vaccinations [14]. Unfortunately, a substantial proportion of healthcare workers show a general lack of intention to get vaccinated, and similar tendencies for vaccine hesitancy were noticed among healthcare workers during the COVID-19 vaccination campaign. Vaccine hesitancy among healthcare workers in Europe has become a source of significant concern in recent years, and it stems from scepticism, lack of confidence, and worries regarding vaccine effectiveness and side effects [15,16]. The study intended to examine the severity of anxiety among healthcare workers during the COVID-19 pandemic, identify any potential links between current anxiety and vaccination attitudes, and investigate whether these variables interact. The study will complement the data concerning the attitudes of Polish healthcare workers towards COVID-19 vaccinations, and also fill the gaps in knowledge about the influence of anxiety on attitudes affecting the environment of healthcare workers. Previous research has provided limited data on the motivational impact of negative emotions on protective actions in COVID-19 prevention.

## 2. Objective of the Work

The aim of the research is the analysis of the level of anxiety among healthcare workers during the COVID-19 pandemic and the assessment of attitudes towards vaccinations against COVID-19.

## 3. Data and Method

### 3.1. Study Design

The cross-sectional study was conducted in 2022 in accordance with the Strengthening the Reporting of Observational Studies in Epidemiology (STROBE) guidelines.

### 3.2. Study Participants

The convenience purposive sampling method was used and the study was led among nurses and doctors employed in healthcare facilities of the Provincial Hospital in Mielec in the Podkarpackie voivodeship in Poland. The inclusion criteria for the research were the participant’s consent to completing the survey, professional activity, i.e., current employment as a nurse or a doctor, and having at least one year of work experience as a healthcare worker. The exclusion criterion was having less than one year of work experience. After obtaining the consent to participate in the study from the subjects, 400 surveys were distributed and 385 responses were received, which means a return rate of 97%. A total of 385 participants who correctly completed the survey were included in the analysis.

### 3.3. Instruments

The Trait Anxiety Scale SL-C and a survey were utilized in the investigation.

The survey contains explicit instructions on how to complete it. It includes open-ended, single- and multiple-choice questions that allow obtaining cadastral, epidemiological, and qualitative information. The survey consists of a general and a detailed part. The general section includes demographic data, such as age, gender, work experience, education, and workplace. The detailed section includes questions about anxiety levels, causes of anxiety, the influence of various factors on anxiety levels, the pandemic’s impact on mental health, the frequency of negative feelings during the pandemic, the use of anxiety-coping strategies during the COVID-19 pandemic, and questions about COVID-19 vaccinations and attitudes towards vaccinations.

The Trait Anxiety Scale SL-C is a tool for measuring the intensity of anxiety as a personality trait, which is understood as an individual’s tendency to perceive and predict future events in the range of dangers or threatening situations, which is manifested through specific symptoms at the emotional, cognitive, somatic, and behavioural level. The SL-C scale is built of 15 items and the answers are given on the 4-point scale from never to often. Every answer given is scored as follows: never—0, rarely—1, sometimes—2, often—3, except for items 9 and 11, where the score was as follows: never—3, rarely—2, sometimes—1, and often—0. The results of the SL-C scale are the sum of all obtained points. The scores range from 0 (lowest degree of anxiety traits) to 45 (highest intensity of anxiety traits).

### 3.4. Data Collection

The prepared research tool was verified, i.e., checked to what extent it measured the phenomenon that we wanted to understand. A pilot study was conducted on a small sample of people to verify and standardize the survey, ensuring that all questions were clear and understandable to respondents, that they were understood in accordance with the researcher’s intention, and that they provided the information the researcher desired. The researcher explained the purpose and meaning of the study to each surveyor and informed them that participation was completely voluntary. Envelopes with information about the study, the informed consent form, and the survey were distributed among all respondents who were asked to return the envelopes after completing them. Doctors and nurses who work in outpatient and surgery departments completed the survey. When the subjects responded to the survey questions, they had no objections. Taking into account refusals and withdrawals from the study, 400 surveys were given during the test phase, with 385 responses, resulting in a 97% return rate. A total of 385 forms (100%) were considered for statistical analysis. The retest took place after an average of one month. In the retest phase, 105 completed surveys were returned. The reduced number of retest participants was largely due to professional absence lasting over two weeks. Due to the high absence level of doctors, this group was not included in the study.

### 3.5. Sample

The study had 385 participants, including 67% women and 33% men. The average age of the patients was SD 48.41 ± 6.76 years.

### 3.6. Ethical Considerations

The study was approved by the Team for Scientific Research Ethics at the Bioethics Committee at Collegium Masoviense University of Health Sciences in Zyrardow and the Institute of Healthcare of the State University of Technology and Economics in Jarosław, no. 197/2022. Participation in the study was voluntary and anonymous and respondents were informed of their right to refuse or withdraw from the study at any time. Each participant was informed of the study’s purpose and deadline.

### 3.7. Data Analysis

The study’s results were statistically summarized and imported into the statistical programme Statistica (version 14.0, TIBCO Software Inc., Palo Alto, CA, USA). Descriptive statistics were used to determine the percentages and 95% confidence interval (CI). Statistical characteristics of continuous variables were presented in the form of arithmetic means, standard deviations, and medians. Statistical characteristics of step and qualitative variables were presented in the form of numerical and percentage distributions using the Student’s *t*-test or Mann–Whitney U test. A correlation was determined using Pearson’s test, while χ^2^ was used for comparison between groups. Significance was assessed at *p* < 0.05. The chi-square test was used to assess the diversity of opinions on vaccination safety in groups. The repeatability of responses to individual questions was assessed using Kappa Cohen statistics. Missing data were excluded from all analyses.

## 4. Results

### 4.1. Demographics Data

The study group included 385 people. The average age of the subjects was 48.41 ± 6.76 years. The nurses constituted 55% of the study group and the doctors 45%. A total of 70% of healthcare workers had over 10 years of work experience. Other descriptive statistics identifying the study group are presented in Table 1.

### 4.2. COVID History

Over half of the subjects (57%, 95% CI: 50–61) suffered from COVID-19. The conducted analysis showed that the incidence of COVID-19 among respondents did not correlate significantly with age, gender, place of residence, position, professional experience, or the department where the respondents worked (Table 2). Table 3 shows that among the study group, fever (50%, 95% CI: 48–53), weariness (46%, 95% CI: 40–49), and loss of taste or smell (38%, 95% CI: 37–51) were the most common symptoms during COVID-19. Of all respondents who suffered from COVID-19, only 25% (95% CI: 20–28) reported a lack of complications following the illness. The most common complications mentioned were impaired concentration (32%, 95% CI: 30–36) and constant feeling of fatigue (28%, 95% CI: 25–30).

### 4.3. Vaccinations against COVID

A total of 85% of respondents received vaccinations (95% CI: 84–89) (Table 4 and Table 5). The mRNA COVID-19 vaccine from Pfizer/BioNTech was the most popular among healthcare workers (79%, 95% CI: 75–81), followed by Johnson & Johnson (13%, 95% CI: 10–16), Moderna (5%, 95% CI: 3–7), and AstraZeneca (3%, 95% CI: 1–6). The analysis showed that the choice of the vaccine type was statistically dependent only on the age of the respondents (*p* = 0.0302). People over 40 were more inclined to choose the Pfizer/BioNTech vaccine, while younger subjects chose Johnson & Johnson more often. The most common source of respondents’ knowledge about the vaccines was peer-reviewed scientific articles (83%, 95% CI: 79–85) (Figure 1). The reasons given by the respondents for choosing the Pfizer/BioNTech vaccine were current availability (33%, 95% CI: 30–36), vaccine composition (28%, 95% CI: 26–30), and lower risk of complications (18%, 95% CI: 15–21). Respondents who chose the Moderna and AstraZeneca vaccines made their choice based on the information about the vaccines’ composition (100%) and effectiveness (100%), while supporters of the Janssen vaccine indicated the possibility of receiving only one dose (45%, 95% CI: 40–51). If they were to get vaccinated, up to 67% (95% CI: 64–69) of unvaccinated respondents would choose Johnson & Johnson’s single-dose vaccination.

More than half of the respondents (52%, 95% CI: 50–57) admitted that they were advised against taking the COVID-19 vaccine by friends (40%, 95% CI: 40–44), co-workers (29%, 95% CI: 24–31), family members (19%, 95% CI: 18–21), and general practitioners (12%, 95% CI: 10–14). The vast majority of the vaccinated respondents (83%, 95% CI: 80–86) decided to get vaccinated on their own, while 16% (95% CI: 15–18) were persuaded to take the vaccine by their co-workers. According to the respondents, the main benefit of vaccination was an increased chance of milder symptoms in case of illness (52%, 95% CI: 50–55). The analysis revealed that reactions to the benefits of getting vaccinated were statistically substantially dependent on employment position (*p* = 0.0142) and professional experience (*p* = 0.0248). The profession is the main factor differentiating the attitudes of medical workers. In their responses, doctors focused mostly on medical statements, stressing that their decision to get vaccinated contributed to achieving population immunity and that vaccines are the most effective method of preventing infections. They also saw it as the only option to terminate the pandemic. However, nurses were more concerned with protecting their loved ones, patients, and themselves. Unlike the doctors, they considered the idea of going to the theatre, the gym, and the cinema, as well as seeing family and friends. The analysis showed a very high correlation of the positive attitude of the surveyed nurses towards vaccinations during the next wave of the pandemic (*p* = 0.0001). The majority of the study group did not support the administration of booster doses (57%, 95% CI: 53–61). The analysis showed that the respondents’ assessment of the validity of the administration of booster doses was very significantly dependent on gender (*p* = 0.0005) and job position of the subjects (*p* = 0.0003). The analysis showed no correlation with the ward where the respondents worked.

A total of 71% (95% CI: 68–76) of respondents believed that vaccines are not harmful. The analysis showed that the respondents’ assessment of the harmfulness of vaccines was statistically significantly dependent on their age (*p* = 0.0409) and job position (*p* = 0.0171), as well as highly statistically dependent on the subject’s gender (*p* = 0.0063). Nurses aged 40–60 years were more likely to have a negative opinion on vaccination against COVID-19. Among the respondents who decided to not receive a COVID-19 vaccine, the main reason was fear of complications after receiving the vaccine (87%, 95% CI: 85–91), concern about vaccine safety (31%, 95% CI: 29–34), and negative opinions of others (25%, 95% CI: 20–28) (Figure 2). As many as 68% of unvaccinated respondents were indifferent towards vaccinations of their loved ones. Almost half of those who have been vaccinated (45%, 280 95% CI: 43–47) do not communicate their opinions with patients and do not advocate or discourage them from getting vaccinated. The same opinion was expressed by 15% (95% CI: 13–19) of unvaccinated respondents. A total of 55% (95% CI: 53–58) of respondents who were vaccinated encouraged their patients to be vaccinated and none of the unvaccinated people declared that they encouraged their patients to be vaccinated. A total of 40% of respondents who received the vaccine encouraged their families and friends to get vaccinated. They declared that 55% (95% CI: 53–58) of their immediate family had been vaccinated. After the analysis, it was found that the vaccination status among the respondents’ closest family members was statistically significantly dependent on the respondents’ gender (*p* = 0.0157), length of service (*p* = 0.0387), and job position (*p* = 0.0130). Doctors were more likely to encourage their family members to get vaccinated. As many as 67% of respondents (95% CI: 65–70) believe that nobody should be forced to get vaccinated. Exactly half of the respondents (50%, 95% CI: 50–56) claim that imposing the obligation of getting vaccinated on medical workers would be a violation of human rights, and 80% (95% CI: 79–83) believe that vaccinations of healthcare workers are not necessary. The analysis showed only a correlation between the respondents’ opinion about obligatory vaccinations of healthcare workers and the ward where the respondents worked. Employees of general medicine wards were more likely to be against mandatory vaccinations (*p* = 0.0130).

### 4.4. Experienced Anxiety

During the COVID-19 pandemic, 68% of the participants (95% CI: 65–70) reported having work-related anxiety regularly, and 34% (95% CI: 33–37) reported having it daily. In self-assessment, the responders usually assessed their anxiety as moderate (50%, 95% CI: 50–56), as mild (33%, 95% CI: 30–36), or strong (17%, 95% CI: 15–19). The most common symptoms of anxiety are unrest (81%, 95% CI: 80–86), difficulty concentrating (80%, 95% CI: 80–86), difficulty performing tasks (67%, 95% CI: 65–69), inability to control emotions (65%, 95% CI: 65–69), and difficulty relaxing (61%, 95% CI: 60–66). Most respondents indicated fear for their family members’ health as the main cause of their anxiety (72%, 95% CI: 70–74) (Table 6). Only 11% (95% CI: 10–15) of respondents sought help from a psychologist due to their anxiety resulting from the COVID-19 pandemic, but 35% of study participants declared their willingness to participate in anti-stress classes. The respondents claimed that the pandemic negatively affected their mental health (44%, 95% CI: 41–48) or that it had no effect on their mental health (52%, 95% CI: 48–57).

The negative emotions most frequently experienced by respondents during the COVID-19 pandemic were fatigue (average score of 2.2 points on a scale from 0 to 3 points), stress (average of 1.83 points), anxiety (average score of 1.7 points), fear (1.55 points), and fear of infection (1.54 points). The respondents were least likely to feel lonely (0.43 points), overwhelmed (0.82 points), or depressed (1.13 points) (Table 7).

The level of anxiety intensity in the subjects was assessed on the SL-C scale. On this scale, the respondents could obtain a maximum of 45 points. The average score was 20,86 points ± 8.39 points (Table 8). A total of 29% (95% CI: 25–33) of the respondents experienced an average level of anxiety, and 22% (95% CI: 20–26) of the respondents had a high level of anxiety (Table 8). It was shown that the level of anxiety intensity based on the SL-C scale varied depending on the gender of the respondents and their profession (*p* = 0.038). Women nurses experienced, on average, a higher level of anxiety than male doctors. The relationship between the level of anxiety intensity and the age of the nursing staff was close to the significance level (*p* = 0.061). People aged less than 40 years experienced a slightly higher level of anxiety than people aged over 40 years. The level of anxiety intensity based on the SL-C scale did not differ depending on the respondents’ professional experience (*p* = 0.117) or workplace (*p* = 0.361). It was also shown that the fact of getting vaccinated against COVID-19 influenced the anxiety intensity level (*p* = 0.033) (Table 9). People who were vaccinated experienced a lower level of anxiety. In correlation analysis using Spearman’s rho coefficient, the relationship between attitude towards the COVID-19 vaccine and anxiety was confirmed. Attitudes towards COVID-19 were positively associated with fear of COVID-19 (rho = 0.27; *p* < 0.001).

The ways of dealing with anxiety varied slightly based on the intensity of anxiety. People who experienced moderate or high levels of anxiety tended to cope with it by listening to music more frequently than those who had low levels of anxiety (*p* = 0.040).

## 5. Discussion

The COVID-19 pandemic is an exceptional experience for healthcare workers who are in constant contact with this virus and infected patients. The introduction of preventive medicine and a significant improvement in sanitary and epidemiological activities can reduce the threat of infectious diseases. Despite scientific and medical consensus on the benefits of vaccination, concerns about vaccine hesitancy are growing. It has a great influence on the attitudes of healthcare workers towards COVID-19 vaccinations, and this study allowed us to learn about them [14,15,17].

This research was conducted on a group of nurses and doctors. After considering various reasons, 85% of respondents decided to get vaccinated against COVID-19. In Solecka’s research, as many as 90% of doctors were vaccinated with 2 doses of the vaccine, and all medical professionals achieved an exemplary result of over 80% [18]. In Cyprus, only 7% of 436 nurses and midwives were hesitant to be vaccinated against COVID-19, while according to a study conducted in France, 23% of 1965 nurses and doctors were hesitant. These numbers suggest very different perceptions among healthcare workers in various locations about the general concerns or concerns about vaccine safety and effectiveness [15]. However, the statistics were not equally optimistic everywhere. The initial percentage of vaccinated healthcare workers in Georgia was only 18%. The situation was better in Italy, where, according to the statement of the Public Health Committee of the Polish Academy of Sciences on vaccinations against COVID-19, 67% of medical workers declared the willingness to get vaccinated after the vaccine was introduced to the market [18,19]. However, studies conducted since March 2022 show that the attitude of Italian medical workers has improved and 91.5% of doctors and nurses received an additional dose of the vaccine [20]. Despite the large number of vaccinated personnel, the vast majority, up to 80%, believe that vaccination is not required to execute the duties of a doctor or nurse, and 50% consider that mandating vaccination obligations on healthcare workers violates human rights. This allegation contradicts various statements, including those made on the Ministry of Health website, which argue that mandating vaccination among healthcare workers is the only way to protect them, citing other EU countries, such as France, Greece, Latvia, and Italy. However, research conducted in these countries shows that local medics, despite being vaccinated, do not agree with the statements regarding compulsory vaccinations, and some of them decide to get vaccinated only because of the possibility of sanctions [21]. Perception of risk, trust, emotions, beliefs, worldviews, controversies, and significant events such as epidemics significantly impact healthcare workers’ attitudes and their perceptions about vaccinations [15].

Collective responsibility appears to be one of the most important motivators for medics who desire to work safely. However, respondents justified their decision to get vaccinated with reasons such as fear of severe illness, protecting their families, or believing that this was the only way to end the COVID-19 pandemic. These arguments seem to be justified, because, as shown by Johnson et al.’s research, the risk of illness and severe symptoms is almost four times higher in unvaccinated people [22]. Cosby noticed that being motivated by collective responsibility is more common among medical workers with a rather negative attitude towards vaccinations [23]. A slightly different approach was observed in a hospital in Munich, where surveyed healthcare workers stated that their motivation was mainly to avoid receiving medication in the treatment of infection or that the overall benefits outweighed the risks associated with vaccination [24]. According to Lou et al., trust is the most important factor that influences the decision to get vaccinated. Due to a lack of trust, the popularization of vaccinations is significantly hampered [25]. Research by Ciesiek-Ślizowska et al. shows a similar tendency [26]. This research also seems to confirm this conclusion, as the surveyed workers express considerable distrust resulting from the rapid introduction of COVID-19 vaccines to the market. Most unvaccinated medics are hesitant to get vaccinated due to the short duration of the study, as shown by Ledda et al.’s research [27].

Many researchers raise the issue of the harmfulness of vaccinations against COVID-19. In our research, 71% of respondents agree that the introduced vaccines are not dangerous, and some of them believe that side effects are inevitable, just as in the case of any other vaccine. Augustynowicz and Jackowska agree with the respondents’ opinion, adding that as the number of vaccinated people increases, the possibility of virus transmission decreases [28]. Opposing attitudes, which are consistent with the opinions of the remaining group of respondents, present numerous arguments related to the harmfulness of the introduced vaccine. These arguments were gathered and described in detail in the research by Ciesek-Ślizowska et al. [26]. Research led by Casuccio and Giuseppe La Torre in China showed that subsequent side effects observed in co-workers and subjects themselves have a significant impact on the negative attitude towards vaccinations. This may considerably contribute to the reluctance to receive booster doses. The most commonly reported side effects were mild or moderate and included headaches and fever [29]. General practitioners participating in the European multi-country study raised worries about insufficient information on vaccine safety and the possibility of overexposure [15].

Working during the COVID-19 pandemic was a difficult situation for healthcare workers, causing negative emotions such as stress or anxiety. Our research shows that 68% of respondents experienced anxiety during the COVID-19 pandemic. Similar results (79%) were obtained by Juszczak [30], which confirms that the pandemic negatively influenced the private lives of medical personnel, causing anxiety of varying intensity. Stress and anxiety are negative emotions that are an inherent part of the COVID-19 pandemic and can vary in intensity. Our research shows that anxiety was experienced in a mild, moderate, or severe form. However, in the research led by Proszek, the respondents reported severe stress and anxiety [31]. Almost half of the respondents experienced anxiety related to the pandemic several times a month, similar to the research by Grzelak and Szwarc, where 60% of the respondents sometimes experienced anxiety [32]. Our research shows that the level of anxiety experienced during the pandemic depends mostly on gender, and, to a lesser extent, on age. Similar results were obtained by Dymecka, who also showed that gender and age influence the level of anxiety [33]. However, in Wilczyńska’s research, no significant relationship was reported between the level of experienced anxiety during the pandemic and age [34]. Our research shows that the main cause of anxiety experienced during the COVID-19 pandemic is fear for the health of family members (72%) and fear for one’s health (60%). Also, Juszczak points out similar causes of anxiety—negative emotions such as fear for one’s health, fear for family members’ health, as well as fear for patients’ health and life [30]. Proszek mentions fear of infection as the main cause of anxiety [31]. Similarly, according to Bluszcz and Matachowska, the main stress factors during the pandemic are fear of getting sick and infecting family members [35]. As shown in our research, ways of coping with anxiety differ slightly from the level of feeling it. In the research conducted by Grzelak and Szwarc, the way of coping with anxiety was not influenced by any of the following factors: age, education, professional experience, or workplace [32]. Medical personnel working during the COVID-19 pandemic are a group that is at high risk of developing mental disorders [36]. A small number of respondents, only 11%, expressed willingness to receive psychological help. Similar results were obtained by Wasik and Koweszko. Their research shows that only 9% of respondents received psychological help during the pandemic [37]. For many years, there has been evidence of a positive impact of physical activity on mental health. Research conducted by a group of scientists from Gothenburg, published in November 2021, shows that physical activity significantly reduces the level of anxiety [38]. Also, according to our research, over half of the respondents (88%) claim that physical activity reduces their anxiety. Among many ways of coping with worry and anxiety, respondents most often chose spending time with family (73%), listening to music (55%), and sport (50%). However, Juszczak showed in her work that the most frequently chosen ways of coping with anxiety and stress are active methods [30].

Our results show that anxiety experienced during the COVID-19 pandemic was positively correlated with attitudes towards vaccinations and was consistent with previous research. Szmyd et al. showed that fear of getting infected with COVID-19 and fear of infecting family members were correlated with the willingness to get vaccinated [39]. Giuliani et al. found that willingness to get vaccinated is highest among people who believe that they may become infected with COVID-19 and become seriously ill [40]. In this context, the results indicating that fear of side effects is a negative correlate of willingness to be vaccinated also seem interesting [39,41]. Similarly, Jach et al. showed that attitudes towards COVID-19 vaccinations were positively related to fear of COVID-19, social distancing, perceived level of susceptibility to illnesses, avoiding germs, attitudes towards science, and knowledge about vaccines. Attitudes towards COVID-19 vaccines were also negatively related to the general tendency toward conspiracy beliefs, including beliefs about COVID-19 being a hoax [42]. The fear of COVID-19 can play a crucial role in shaping public messages about the COVID-19 vaccine. Too low a level of fear of COVID-19 may lead to a lower level of preventive behaviours and readiness to get vaccinated. From a different perspective, trust in science, scientists, and political leaders is crucial for effective large-scale preventive actions, including vaccination programs. [42,43].

It is also important to mention that there is evidence that advice from healthcare workers impacts patients’ decisions about vaccination. The role of healthcare workers has been identified as an appropriate category of determinants of vaccine hesitancy, as healthcare workers can provide their patients with personalized information and recommendations and have a positive impact on vaccine uptake [44,45]. In a study conducted by Charmasson et al., 65.2% of patients talked to a healthcare worker about COVID-19 vaccines, and 55% of patients were recommended to be vaccinated by their healthcare professional. The majority of the conversations were with a general practitioner (48.9%), and according to 52.5% of participants, the discussion had a positive impact [44]. These findings complement the results of a large study conducted by Nguyen et al. that shows that recommending COVID-19 vaccines by healthcare professionals influenced patients’ perception of the effectiveness and safety of COVID-19 vaccines. Nguyen et al. also found that recommendations of healthcare professionals had an objective effect on vaccination absorption. These results indicate that conversations with healthcare professionals may influence patients [46]. Lasagna et al. showed that during the first oncological visit, only 30% of doctors usually propose a vaccination at the first visit, 41% usually do not discuss vaccinations during the first visit, and 29% recommend vaccinations only to specific categories of patients. According to 56% of respondents, patients are more aware of the benefits of vaccines, while 36% stated that patients were concerned about receiving too many vaccines [45]. In our research, 60% of healthcare professionals did not share their views with patients, nor did they encourage or discourage them from getting vaccinated. A total of 55% of respondents who were vaccinated against COVID-19 encouraged their patients to get vaccinated. A total of 40% of vaccinated respondents encourage their families and friends to get vaccinated. They declared that 55% of their immediate family had been vaccinated.

To summarize, the pandemic had a significant impact not only on physical but also on mental health, resulting in a variety of physical and mental burdens. The COVID-19 pandemic, which has the characteristics of a catastrophic disaster, created a difficult situation that caused fear among medical personnel. However, the conducted research shows that despite differing opinions about vaccinations themselves, there is a large percentage of vaccinated healthcare workers, which has significantly reduced the sense of fear and anxiety. Moreover, a positive attitude of doctors and nurses towards vaccinations increases engagement in the vaccination process and has a measurable impact on the promotion of vaccinations not only among their patients but also in society.

## 6. Limitations and Strengths of the Study

Limitations of this study include the small size of the sample, which consisted of volunteers and may not have been representative of the general population. Other weaknesses of the study include its cross-sectional nature and lack of analysis of the work environment and the impact of public health strategies. We did not take into account determinants of anxiety linked with existing comorbidities, high-risk exposure, social responsibilities, and mental vulnerability. The presented study also has strengths, including using a reliable, validated tool to measure anxiety. Another strength of the study is that it highlights psychological variables associated with attitudes toward the COVID-19 vaccine, which may determine attitudes toward vaccines more generally, or even general medical or scientific recommendations. In the face of surprising results regarding psychological risks, the authors will use a validated tool for assessing anxiety and depression in their future research, which will facilitate the conclusion-making process. The comparative study will allow us to obtain information on how time and level of control over the pandemic influenced the level of vaccination and the current mental state of medics.

## 7. Conclusions

Almost all of the questioned doctors chose to get vaccinated, but nurses had a substantially lower percentage of vaccinated people, implying that work status has a considerable impact on the decision to get vaccinated against COVID-19. The profession is the most important factor in determining attitudes towards vaccinations against COVID-19. In their answers, doctors stressed that their decision to get vaccinated contributed to achieving population immunity and that vaccines are the most effective method of preventing infections and the only way to terminate the pandemic. The nurses, however, were more concerned with protecting their loved ones, patients, and themselves. Unlike the doctors, they considered the possibility of vacation, going to the gym, cinema, and theatre, as well as meeting family and friends.The gender of the respondents differed greatly in their attitudes about immunisations. Women were far more likely than men to declare that they got vaccinated because they did not want to endanger their families and patients; they wanted to quit wearing protective masks and return to normality. Men stated that vaccines are considered the most effective method of protecting against infection with various diseases, historically used to combat many infectious diseases and that it is the only possibility to end the COVID-19 pandemic. Other factors slightly differentiated the attitudes of medical workers towards vaccinations against COVID-19.The vast majority of medical workers, who based their expertise on peer-reviewed studies, opted to get vaccinated. They justified their decision primarily by increasing their chances of a milder course of the disease if they contracted it, which is consistent with the majority of the research carried out thus far, confirming the hypothesis that medical workers’ knowledge of COVID-19 vaccinations influences their decisions to get vaccinated.In self-assessment, during the COVID-19 epidemic, medical workers often experienced moderate anxiety. On average, women felt more anxious than males. Younger people had slightly higher levels of anxiousness.

## Figures and Tables

**Figure 1 vaccines-12-00366-f001:**
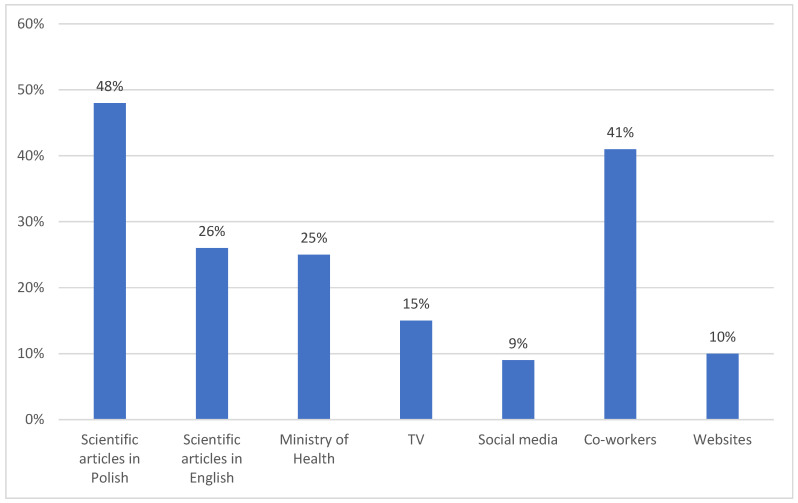
Sources of respondents’ knowledge about vaccines.

**Figure 2 vaccines-12-00366-f002:**
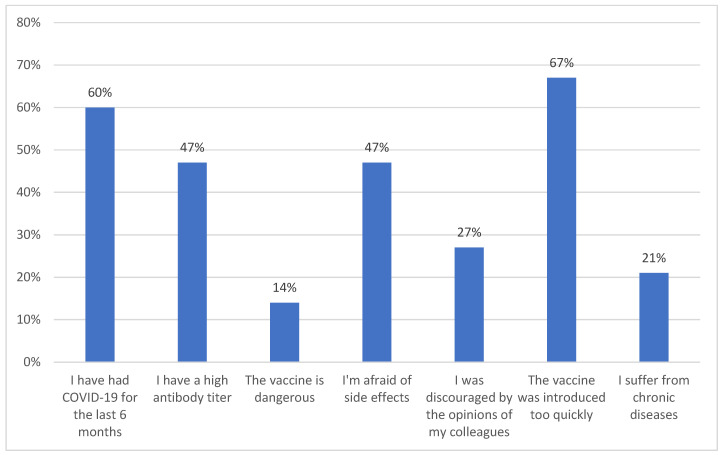
Analysis of the reasons influencing respondents’ lack of decision to get vaccinated.

**Table 1 vaccines-12-00366-t001:** Descriptive statistics of the examined group.

Demographic Information	Test N = 385	Retest N = 105	*p*
Characteristics % (N)			
Sex			
women	67% (258)	77% (81)	0.01
men	33% (127)	23% (24)	
The age of the study group			
SD	48.41 (6.76)	39.31 (5.79)	0.12
95%CI	<25; 60>	<25; 60>	
Place of residence			
city	65% (250)	75% (79)	0.22
village	35% (135)	25% (26)	
Financial situation			
very good	13% (50)	20% (21)	0.12
good	58% (223)	68% (71)	
average	18% (69)	10% (11)	
bad	10% (39)	1% (1)	
very bad	1% (4)	1% (1)	
Age groups			
20–30	14% (54)	25% (26)	0.01
31–40	23% (89)	20% (21)	
41–50	25% (96)	36% (38)	
51–60	38% (146)	19% (20)	
Profession			
nurses	55% (212)	75% (79)	0.01
doctors	45% (173)	25% (26)	
Marital status			
married	74% (285)	80% (84)	0.61
widowed	2% (8)	0% (0)	
unmarried	24% (92)	20% (21)	
Education of the study group			
lack of specialization	46% (177)	21% (22)	0.60
having a specialization	54% (208)	79% (83)	
Seniority			
up to 10 years	30% (116)	41% (43)	0.01
11–20 years	25% (96)	20% (21)	
21–30 years	25% (96)	25% (26)	
over 30 years	20% (77)	14% (15)	
Workplace			
surgical department	32% (123)	35% (37)	0.71
general department	26% (100)	43% (45)	
hospital emergency ward	25% (96)	15% (16)	
intensive care unit	17% (66)	7% (7)	

**Table 2 vaccines-12-00366-t002:** Analysis of the correlation between respondents’ incidence of COVID-19 and selected variables.

Result Student’s *t*-Test	Variables					
	Age	Sex	Place of Residence	Profession	Seniority	Workplace
**Pearson correlation coefficient**	−0.1259	−0.0124	0.1005	−0.0048	−0.1362	0.1097
** *p* **	0.3372	0.9023	0.3198	0.9626	0.6485	0.9915

**Table 3 vaccines-12-00366-t003:** Symptoms accompanying respondents while suffering from COVID-19.

Symptoms	Sex		*p*	Profession		*p*
	Women	Men		Nurses	Doctors	
Characteristics % (N)						
Fever	50% (129)	39% (50)	0.27	55% (117)	59% (102)	0.25
Cough	32% (83)	25% (32)	0.66	36% (76)	29% (50)	0.55
Fatigue	46% (119)	42% (53)	0.17	49% (104)	52% (90)	0.99
Loss of taste or smell	38% (98)	31% (39)	0.41	40% (85)	39% (67)	0.21
Sore throat	10% (26)	17% (22)	0.55	12% (25)	19% (33)	0.41
Headache	29% (78)	11% (14)	0.41	32% (68)	15% (26)	0.55
Muscle pain	25% (65)	13% (17)	0.71	30% (64)	23% (40)	0.88
Skin rash	2% (5)	13% (17)	0.91	10% (21)	19% (33)	0.74
Red eyes	4% (10)	18% (23)	0.88	8% (17)	15% (26)	0.53
Breathing difficulties	2% (5)	5% (6)	0.54	7% (15)	8% (14)	0.16
Confusion	0% (0)	10% (13)	0.91	0% (0)	1% (2)	0.16
Pain in the chest	11% (28)	18% (23)	0.44	15% (32)	20% (35)	0.32
No symptoms	7% (18)	19% (24)	0.91	17% (36)	10% (17)	0.74

**Table 4 vaccines-12-00366-t004:** Analysis of the correlation between the respondents’ decision to vaccinate and selected variables.

Result Student’s *t*-Test	Variables					
	Age	Sex	Place of Residence	Profession	Seniority	Workplace
**Pearson correlation coefficient**	−0.0381	−0.0845	0.1227	0.0571	0.0246	−0.1697
** *p* **	0.6204	0.4030	0.2238	0.5729	0.6301	0.2779

**Table 5 vaccines-12-00366-t005:** Symptoms experienced by respondents after receiving the COVID-19 vaccine.

Symptoms	Sex		*p*	Profession		*p*
	Women	Men		Nurses	Doctors	
Characteristics % (N)						
Pain at the injection site	53% (137)	49% (62)	0.27	59% (125)	47% (81)	0.25
Diarrhoea	3% (8)	15% (19)	0.17	5% (11)	17% (29)	0.55
Fatigue	47% (121)	52% (66)	0.66	57% (120)	62% (107)	0.89
Chills	20% (52)	35% (44)	0.41	27% (57)	30% (52)	0.88
Fever	25% (65)	27% (34)	0.55	35% (74)	37% (64)	0.41
Headache	57% (147)	61% (77)	0.41	62% (131)	61% (106)	0.55
Muscle pain	18% (46)	33% (42)	0.71	20% (42)	31% (54)	0.66
Weakness	7% (18)	17% (22)	0.21	10% (21)	12% (21)	0.23
Difficulty breathing	4% (10)	10% (13)	0.14	3% (6)	7% (12)	0.16
No symptoms	24% (62)	18% (23)	0.44	21% (45)	20% (35)	0.22

**Table 6 vaccines-12-00366-t006:** Causes of anxiety during the COVID-19 pandemic.

Causes of Anxiety	Sex		*p*	Profession		*p*
	Women	Men		Nurses	Doctors	
Characteristics % (N)						
Fear for your own health	60% (155)	41% (52)	0.41	66% (140)	51% (88)	0.25
Fear for the health of loved ones	72% (186)	53% (67)	0.37	67% (142)	57% (99)	0.55
The need to remain in isolation	17% (44)	11% (14)	0.91	37% (78)	9% (16)	0.01
The need to care for infected patients	41% (106)	20% (25)	0.18	51% (108)	15% (26)	0.01
Controversies regarding vaccinations against COVID-19	38% (98)	18% (23)	0.59	41% (87)	22% (38)	0.41
Constant information in the media about the number of infections and deaths	52% (134)	25% (32)	0.21	59% (125)	36% (62)	0.55

**Table 7 vaccines-12-00366-t007:** Negative emotions experienced during the COVID-19 pandemic.

Negative Emotions	At All 0	Rarely 1	Often 2	Very Often 3	He Doesn’t Know	M
Characteristics % (N)						
Stress	34% (130)	28% (108)	12% (46)	3% (12)	23% (89)	1.83
Bow	16% (62)	34% (130)	24% (92)	23% (89)	3% (12)	1.55
Anxiety	13% (50)	28% (108)	31% (120)	24% (92)	4% (15)	1.70
Fear of infection	16% (62)	32% (123)	25% (96)	21% (81)	6% (23)	1.54
Tiredness	6% (23)	9% (35)	41% (158)	40% (154)	4% (15)	2.20
Depression	24% (92)	33% (127)	23% (89)	7% (27)	13% (50)	1.13
Nervousness	4% (15)	46% (177)	32% (123)	6% (23)	12% (46)	1.43
Irritability	17% (66)	33% (127)	27% (104)	6% (23)	17% (65)	1.28
Loneliness	60% (231)	12% (46)	7% (27)	3% (12)	18% (69)	0.43
Overwhelmed	34% (130)	28% (108)	12% (46)	3% (12)	23% (89)	0.82

**Table 8 vaccines-12-00366-t008:** Level of anxiety severity based on the SL-C scale.

SL-C	Basic Descriptive Statistics							
	N	AVG	M	Min.	Max.	Q I	Q III	SD
**(0-45 points)**	385	20.9	21.0	2.0	39.0	14.5	28.0	8.4

**Table 9 vaccines-12-00366-t009:** Level of anxiety severity based on the SL-C scale depending on the variables.

Level of Anxiety Severity	Seniority		Profession		Age		Sex	
	Up to 20 Years	Over 20 Years	Nurses	Doctors	Up to 40 Years	Over 40 Years	Women	Men
Low level	41% (87)	57% (99)	40% (85)	78% (135)	38%(54)	58% (140)	44% (114)	74% (94)
Medium level	36% (76)	20% (35)	37% (78)	19% (33)	38% (54)	21% (51)	30% (77)	21% (27)
High level	23% (49)	23% (39)	23% (49)	3% (5)	24% (35)	21% (51)	26% (67)	5% (6)
	**χ²(2) = 4.28 *p* = 0.117**		**χ²(2) = 6.49 *p* = 0.038**		**χ²(2) = 5.60 *p* = 0.061**		**χ²(2) = 6.49 *p* = 0.038**	

## Data Availability

The data are not publicly available due to privacy and ethical restrictions. The data presented in this study may be available conditionally from the corresponding author.

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
