# Peer review of "Anxiety Levels among Healthcare Workers during the COVID-19 Pandemic and Attitudes towards COVID-19 Vaccines"

_vaccines, 2024, doi:10.3390/vaccines12040366_

Round 1

Reviewer 1 Report

Comments and Suggestions for Authors

The paper is very interesting, but some points need to be implemented.

Please check the figures (Figure 1 is probably missing, while Figure 2 has been reported twice).

Is it possible to establish a correlation based on the department of origin (surgery versus internal medicine?).

Discuss for example: PMID: 38005989

How did these health care providers behave toward the patients? (Did they suggest vaccinations? Did those who did not receive the vaccinations tend to propose them to their patients?)

Discuss for example: PMID: 38346925, PMID: 37586016

Comments on the Quality of English Language

Minor editing of the English language required

Author Response

Dear Reviewer,

I would like to thank for their comments. After analysing all the comments, I made the following changes:

  • Please check the figures (Figure 1 is probably missing, while Figure 2 has been reported twice)

Missing figure 1 added (line 236)

  • Is it possible to establish a correlation based on the department of origin (surgery versus internal medicine?).

In each section of the results, the occurrence of correlations between gender, age, job position, seniority and workplace was assessed. The research did not show any correlation between departments (results section)

  • Discuss for example: PMID: 38005989

This item was not added due to the very high citation of Vaccines

  • How did these health care providers behave toward the patients? (Did they suggest vaccinations? Did those who did not receive the vaccinations tend to propose them to their patients?)

Employee attitudes are included in the results section (line 219-291)

  • Discuss for example: PMID: 38346925, PMID: 37586016

These references have been added (item 44 and 45)

I hope that the changes made are satisfactory and this will allow publication. I am asking you to take into account the positive comments of the reviewers that this is an interesting study and a good study.

Sincerely

Authors

Reviewer 2 Report

Comments and Suggestions for Authors

REVIEWER'S REPORT

Manucsript title: Anxiety Levels among Healthcare Workers during the 2 COVID-19 Pandemic and Attitudes Towards COVID-19 (Authors: Lewandowska et al.).

   In this study, the authors used a statistical approach to investigate the intensity of anxiety among healthcare workers during the COVID-19 pandemic, identify any possible correlations between their existing anxiety and attitude towards vaccination, and determine whether these variables had an impact on each other.  

  Since cross-sectional analysis was done in 2022, the question arises as to whether this work is still relevant.

  First and foremost, even if the English is pretty clear, I suggested that the manuscript's text be corrected and/or paraphrased in many places, as this would greatly improve its quality.

Scientific titles (e.g., professor, PhD, etc.) preceding the authors' names must be deleted.

In Abstract. The sentence in line 29-30, should be paraphrased as follows "The cross-sectional study followed the recommendations of STROBE (Strengthening the Reporting of Observational Studies in Epidemiology)." In lines 32-34, the sentences should be written as "The study included 385 participants, with an average age of 48.41±6.76 years." The sentence in lines 35-36 should be paraphrased as "85% of respondents have received vaccination. 71% of respondents believe vaccinations are harmless." Frequently, the participants assessed their level of anxiety as moderate." In lines 38-44, the sentences should be paraphrased as follows "Almost all surveyed doctors chose to be vaccinated, while the percentage of vaccinated nurses was significantly lower. As a result, it is possible to conclude that the employment position has a significant influence on the decision to get vaccinated against COVID-19. During the COVID-19 pandemic, most healthcare professionals experienced moderate levels of anxiety. During the COVID-19 pandemic, most healthcare professionals experienced moderate levels of anxiety. Receiving the COVID-19 vaccination reduced the level of anxiety."

In Introduction. The sentences in lines 48-52 should be rewritten as follows "The psychological impact of the COVID-19 pandemic on individual, group, and societal health is a multidimensional topic. It is the focus of current in-depth research efforts.

The psychological impact of the COVID-19 epidemic on the health of individuals, groups, and society is complicated and multifaceted. It is the focus of ongoing in-depth scientific investigations. What is essential is that, given the pandemic's potential long-term psychological impact, a comprehensive assessment of the event will not be feasible until many years have passed." In line 55, "Undoubtedly..." should be replaced by "Without a doubt..." For sentence in lines 68-69, please provide the reference.  The sentence in lines 78-83 should be rewritten as "During the pandemic, healthcare personnel faced heightened stress and chronic anxiety. They put not just their own lives and health at risk while carrying out their duty, but also the lives of others they care about. Healthcare personnel face daily challenges such as concern of infecting their families, a lack of adequate personal protective equipment, and work stress." In lines 84-85, the sentence should be slightly paraphrased as "Considerations to date reveal how devastating the psychological implications of the pandemic are for the social, economic, and healthcare systems." The sentence in lines 88-91 should be rewritten as "There are numerous recommendations for implementing possible activities at each level stated, but vaccination is the common action for all of them." The sentences in lines 90-91, the sentences should be written as "Vaccines serve a highly vital role in health, particularly in preventing against infectious diseases such as COVID-19."

The sentences in lines 94-96 should be written as "Despite the fact that vaccinations are proven to be successful, there are still differing perspectives and theories about this approach. Opinions differ not only among the general public, but even among doctors." The sentences in lines 103-110, should be paraphrased as "Vaccine hesitancy among healthcare workers in Europe has become a source of significant concern in recent years, and it stems from scepticism, lack of confidence, and worries regarding vaccine effectiveness and side effects. The study intended to examine the severity of anxiety among healthcare workers during the COVID-19 pandemic, identify any potential links between current anxiety and vaccination attitudes, and investigate whether these variables interacted."

 In subsection 3.3, Instruments. The sentence in lines 130-131, should be written as "The Trait Anxiety Scale SL-C and a survey were utilized in the investigation. The survey contains explicit instructions on how to complete it." The sentence on lines 135-140 I would advise paraphrasing as follows "The detailed section includes questions about anxiety levels, causes of anxiety, the influence of various factors on anxiety levels, the pandemic's impact on mental health, the frequency of negative feelings during the pandemic, the use of anxiety-coping strategies during the COVID-19 pandemic, and questions about COVID-19 vaccinations and attitudes towards vaccinations." In sentence on line 141 "...which allows..." should be replaced by "...for...".  The sentence on lines 149-150 should be written as " The scores range from 0 (lowest degree of anxiety traits) to 45 (highest intensity of anxiety traits).

 In subsection 3.4, Data collection. The sentence on lines 154-158, should be paraphrased in certain places as follows " A pilot study was conducted on a small sample of people to verify and standardize the survey, ensuring that all questions were clear and understandable to respondents, that they were understood in accordance with the researcher's intention, and that they provided the information the researcher desired." In lines 162-165, the sentences should be paraphrased as " Doctors and nurses who work in outpatient and surgery departments completed the survey. When the subjects responded to the survey questions, they had no objections."  The sentence in lines 167-170 should be written as follows " Taking into account refusals and withdrawals from the study, 400 surveys were given during the test phase, with 385 responses, resulting in a 97% return rate. 385 forms (100%) were considered for statistical analysis. The retest was conducted after an average of one month."

In subsection 3.5. Sample, the sentence in lines 174-175 should be paraphrased and corrected as follows " The study had 385 participants, including 67% women and 33% men. The average age of the patients was SD 48.41 ± 6.76 years."

 In subsection 3.6. The sentence in lines 181-182 should be paraphrased as follows "Each participant was informed of the study's purpose and deadline."

 In subsection 3.7, Data analysis. The sentence on line 184-185 should be written as follows " The study's results were statistically summarized and imported into the statistical programme Statistica (version 14.0,TIBCO Software Inc.). The sentence on lines 193-197 should be paraphrased as " Because the distributions of the majority of the analysed quantitative variables did not match the normality criteria, non-parametric Mann-Whitney U coefficient and Spearman's rho correlation coefficient tests were used in the analyses. Another factor utilized to describe the relationship was the Pearson correlation coefficient rxy." The sentence on lines 200-204 should be written as follows " The higher  rxy coefficient, the stronger the correlation between the data, for which the following levels are assumed: rxy = 0 (no correlation), 0<rxy <0.1 (slight correlation), 0.1<rxy <0.3 (weak correlation), 0.3<rxy <0.5 (average correlation), 0.5<rxy <0.7 (strong correlation), 0.7<rxy <0.9 (very high correlation), and 0.9<rxy <1 (nearly full correlation)".Please provide references for statistical approaches utilized in this work. It's not completely clear whether the Pearson correlation coefficient (r) or its square value were used to assess correlations.  

In Results. Subsection 4.2. The sentence on lines 216-218 should be paraphrased as " Table 3 shows that among the study group, fever (50%, 95% CI: 48–51), weariness (46%, 95% CI: 40–49), and loss of taste or smell (38%, 95% CI: 37–51) were the most common symptoms during COVID-19."

 In subsection 4.3. The sentence on line 227 should be slightly corrected and written as " 85% of respondents received vaccinations (95% CI: 84-89) (Tables 4 and 5)." The sentence on lines 241-242, should be written as " If they were to get vaccinated, up to 67% (95% CI: 64-69) of unvaccinated respondents would choose Johnson & Johnson's single-dose vaccination." The sentence in lines 255-257 should be written as " The analysis revealed that reactions to the benefits of getting vaccinated were statistically substantially dependent on employment position (p=0.0142) and professional experience (p=0.0248)."  The sentences in lines 260-264 should be paraphrased as follows " They also saw it as the only option to terminate the pandemic. However, nurses were more concerned with protecting their loved ones, patients, and themselves. Unlike the doctors, they considered the idea of going to the theatre, the gym, and the cinema as well as seeing family and friends." The sentence in lines 280-282 should be written as " Almost half of those who have been vaccinated (45%, 280 95% CI: 43-47) do not communicate their opinions with patients and do not advocate or discourage them from getting vaccinated."

 In subsection 4.4. The sentence in lines 302-303 should be paraphrased as " During the COVID-19 pandemic, 68% of the participants (95% CI: 65-70) reported having work-related anxiety regularly, and 34% (95% CI: 33-35) reported having it daily." The sentence in lines 341-343 should be corrected and paraphrased as " The ways of dealing with anxiety varied slightly based on the intensity of anxiety. People who experienced moderate or high levels of anxiety tended to cope with it by listening to music more frequently than those who had low levels of anxiety (p=0.040)."

In Discussion. In lines 346-348, the sentence should be written as " Infectious diseases, including SARS-CoV-2, have been and continue to be a major social problem, but more importantly, a health problem - both physically and mentally. Even if we do not become infected, we are nevertheless exposed to the psychological effects of the COVID-19 pandemic. The emergence of severe infectious diseases causes increased anxiety and fear throughout society, as seen by earlier MERS, H1N1 and SARS epidemics. As a result of the increased number of infections and deaths caused by infections, society is terrified of death, infected people and infection itself. All these circumstances led to the elevated levels of anxiety and fear." The sentence in lines 362-363 should be written as follows " After considering various reasons, 85% of respondents decided to get vaccinated against COVID-19." The sentences in lines 377-380, can be written as follows " Despite the large number of vaccinated personnel, the vast majority, up to 80%, believe that vaccination is not required to execute the duties of a doctor or nurse, and 50% consider that mandating vaccination obligations on healthcare workers violates human rights."  The sentence in line 383-383, should be paraphrased as "This allegations condradicts various statements, including those made on the Ministry of Health website, which argue that mandating vaccination among healthcare workers is the only way to protect them, citing other EU countries, such as France, Greece, Latvia and Italy." The sentence in line 390-393 should be paraphrased as "Collective responsibility appears to be one of the most important motivators for medics who desire to work safely. However, respondents justified their decision to get vaccinated with reasons such as fear of severe illness, protecting their families, or believing that this was the only way to end the COVID-19 pandemic." The sentence in lines 408-410 should be paraphrased as "Most unvaccinated medics are hesitant to get vaccinated due to the short duration of the study, as shown by research by Ledda et al." The sentence in lines 424-426 should be written as "General practitioners participating in the European multi-country study raised worries about insufficient information on vaccine safety and the possibility of overexposure."

The statement in lines 478-480 does not make it obvious what it is trying to say.

The sentence in lines 511-512 should be written as "To summarize, the pandemic had a significant impact not only on physical but on mental health, resulting in a variety of physical and mental burdens. The COVID-19 pandemic, which has the characteristics of a catastrophic disaster, created a difficult situation that caused fear among medical personnel."

 In  Conclusion (section 7). The sentences in lines 530-540 should be written as "Almost all of the questioned doctors chose to get vaccinated, but nurses had a substantially lower percentage of vaccinated people, implying that work status has a considerable impact on the decision to get vaccinated against COVID-19. The profession is the most imporatnt factor in determining attitudes towards vaccinations against COVID-19. In their answers, doctors focused mostly on medical statements, stressing that their decision to get vaccinated contributed to achieving population immunity and that vaccines are the most effective method of preventing infections. They also believed that it was the only way to terminate the pandemic. The nurses, however, were more concerned with  protecting their loved ones, patients and themselves. Unlike the doctors, they considered the possibility of vacation, going to the gym, cinema, theatre, as well as meeting family and friends".  The sentences in lines 541-544 should be written as follows "The gender of the respondents differed greatly in their attitudes about immunisations. Women were far more likely than men to declare that they got vaccinated because they didn't want to endanger their families and patients; they wanted to quit wearing protective masks and return to normality." The sentence in lines 550-555 should be paraphrased as follows "The vast majority of medical workers, who based their expertise on peer-reviewed studies, opted to get vaccinated. They justified their decision primarily by increasing their chances of a milder course of the disease if they contracted it, which is consistent with the majority of the research done thus far, confirming the hypothesis that medical workers' knowledge of COVID-19 vaccinations influences their decisions to get vaccinated." The sentences in lines 556-558 should be paraphrased as "During the COVID-19 epidemic, medical workers often experienced moderate anxiety. On average, women felt more anxious than males. Younger people had slightly higher levels of anxiousness."

 It appears to me that discussion (particularly the first half) may be condensed/shortened to make it more appealing to readers. The same should be done with the conclusions.

 The error bars in the histograms in Figures 1 and 2 are required. In Tables and in several places throughout the manuscript's text, commas in numbers must be replaced by dots.

    When viewed as a whole, this work does not leave a strong impression, and its relevance is ambiguous. It might, however, be accepted for publication after extensive editing.

Comments on the Quality of English Language

 I suggested that the manuscript's text be revised and/or paraphrased in numerous places.

Author Response

Dear Reviewer,

I would like to thank for their comments. After analysing all the comments, I made the following changes:

  • First and foremost, even if the English is pretty clear, I suggested that the manuscript's text be corrected and/or paraphrased in many places, as this would greatly improve its quality.

The manuscript has been linguistically corrected

  • Scientific titles (e.g., professor, PhD, etc.) preceding the authors' names must be deleted.

Deleted (line 5-6)

  • In Abstract. The sentence in line 29-30, should be paraphrased as follows "The cross-sectional study followed the recommendations of STROBE (Strengthening the Reporting of Observational Studies in Epidemiology)." In lines 32-34, the sentences should be written as "The study included 385 participants, with an average age of 48.41±6.76 years." The sentence in lines 35-36 should be paraphrased as "85% of respondents have received vaccination. 71% of respondents believe vaccinations are harmless." Frequently, the participants assessed their level of anxiety as moderate." In lines 38-44, the sentences should be paraphrased as follows "Almost all surveyed doctors chose to be vaccinated, while the percentage of vaccinated nurses was significantly lower. As a result, it is possible to conclude that the employment position has a significant influence on the decision to get vaccinated against COVID-19. During the COVID-19 pandemic, most healthcare professionals experienced moderate levels of anxiety. During the COVID-19 pandemic, most healthcare professionals experienced moderate levels of anxiety.Receiving the COVID-19 vaccination reduced the level of anxiety."

Changed as recommended (line 22-42)

  • In Introduction. The sentences in lines 48-52 should be rewritten as follows "The psychological impact of the COVID-19 pandemic on individual, group, and societal health is a multidimensional topic. It is the focus of current in-depth research efforts. The psychological impact of the COVID-19 epidemic on the health of individuals, groups, and society is complicated and multifaceted. It is the focus of ongoing in-depth scientific investigations. What is essential is that, given the pandemic's potential long-term psychological impact, a comprehensive assessment of the event will not be feasible until many years have passed." In line 55, "Undoubtedly..." should be replaced by "Without a doubt..." For sentence in lines 68-69, please provide the reference.  The sentence in lines 78-83 should be rewritten as "During the pandemic, healthcare personnel faced heightened stress and chronic anxiety. They put not just their own lives and health at risk while carrying out their duty, but also the lives of others they care about. Healthcare personnel face daily challenges such as concern of infecting their families, a lack of adequate personal protective equipment, and work stress." In lines 84-85, the sentence should be slightly paraphrased as "Considerations to date reveal how devastating the psychological implications of the pandemic are for the social, economic, and healthcare systems." The sentence in lines 88-91 should be rewritten as "There are numerous recommendations for implementing possible activities at each level stated, but vaccination is the common action for all of them." The sentences in lines 90-91, the sentences should be written as "Vaccines serve a highly vital role in health, particularly in preventing against infectious diseases such as COVID-19." The sentences in lines 94-96 should be written as "Despite the fact that vaccinations are proven to be successful, there are still differing perspectives and theories about this approach. Opinions differ not only among the general public, but even among doctors." The sentences in lines 103-110, should be paraphrased as "Vaccine hesitancy among healthcare workers in Europe has become a source of significant concern in recent years, and it stems from scepticism, lack of confidence, and worries regarding vaccine effectiveness and side effects. The study intended to examine the severity of anxiety among healthcare workers during the COVID-19 pandemic, identify any potential links between current anxiety and vaccination attitudes, and investigate whether these variables interacted."

The introduction was slightly shortened, some content from the discussion was moved to strengthen the purpose of the research, and information about knowledge gaps was added.

All proposals to change the wording have been implemented. (line 46-119)

  • In subsection 3.3, Instruments. The sentence in lines 130-131, should be written as "The Trait Anxiety Scale SL-C and a survey were utilized in the investigation. The survey contains explicit instructions on how to complete it." The sentence on lines 135-140 I would advise paraphrasing as follows "The detailed section includes questions about anxiety levels, causes of anxiety, the influence of various factors on anxiety levels, the pandemic's impact on mental health, the frequency of negative feelings during the pandemic, the use of anxiety-coping strategies during the COVID-19 pandemic, and questions about COVID-19 vaccinations and attitudes towards vaccinations." In sentence on line 141 "...which allows..." should be replaced by "...for...".  The sentence on lines 149-150 should be written as " The scores range from 0 (lowest degree of anxiety traits) to 45 (highest intensity of anxiety traits).

All proposals to change the wording of the sentences have been implemented (line 139-157)

  • In subsection 3.4, Data collection. The sentence on lines 154-158, should be paraphrased in certain places as follows " A pilot study was conducted on a small sample of people to verify and standardize the survey, ensuring that all questions were clear and understandable to respondents, that they were understood in accordance with the researcher's intention, and that they provided the information the researcher desired." In lines 162-165, the sentences should be paraphrased as " Doctors and nurses who work in outpatient and surgery departments completed the survey. When the subjects responded to the survey questions, they had no objections."  The sentence in lines 167-170 should be written as follows " Taking into account refusals and withdrawals from the study, 400 surveys were given during the test phase, with 385 responses, resulting in a 97% return rate. 385 forms (100%) were considered for statistical analysis. The retest was conducted after an average of one month."

All proposals to change the wording of the sentences have been implemented (line 159-175)

  • In subsection 3.5. Sample, the sentence in lines 174-175 should be paraphrased and corrected as follows " The study had 385 participants, including 67% women and 33% men. The average age of the patients was SD 48.41 ± 6.76 years."

All proposals to change the wording of the sentences have been implemented (line 177-178)

  • In subsection 3.6. The sentence in lines 181-182 should be paraphrased as follows "Each participant was informed of the study's purpose and deadline."

All proposals to change the wording of the sentences have been implemented (line 180-185)

  • In subsection 3.7, Data analysis. The sentence on line 184-185 should be written as follows " The study's results were statistically summarized and imported into the statistical programme Statistica (version 14.0,TIBCO Software Inc.). The sentence on lines 193-197 should be paraphrased as " Because the distributions of the majority of the analysed quantitative variables did not match the normality criteria, non-parametric Mann-Whitney U coefficient and Spearman's rho correlation coefficient tests were used in the analyses. Another factor utilized to describe the relationship was the Pearson correlation coefficient rxy." The sentence on lines 200-204 should be written as follows " The higher  rxy coefficient, the stronger the correlation between the data, for which the following levels are assumed: rxy = 0 (no correlation), 0<rxy <0.1 (slight correlation), 0.1<rxy <0.3 (weak correlation), 0.3<rxy <0.5 (average correlation), 0.5<rxy <0.7 (strong correlation), 0.7<rxy <0.9 (very high correlation), and 0.9<rxy <1 (nearly full correlation)".Please provide references for statistical approaches utilized in this work. It's not completely clear whether the Pearson correlation coefficient (r) or its square value were used to assess correlations.  

The provision regarding statistical analysis has been clarified (line 187-197)

  • In Results. Subsection 4.2. The sentence on lines 216-218 should be paraphrased as "Table 3 shows that among the study group, fever (50%, 95% CI: 48–51), weariness (46%, 95% CI: 40–49), and loss of taste or smell (38%, 95% CI: 37–51) were the most common symptoms during COVID-19."

The sentence has been changed (line 206-215)

  • In subsection 4.3. The sentence on line 227 should be slightly corrected and written as " 85% of respondents received vaccinations (95% CI: 84-89) (Tables 4 and 5)." The sentence on lines 241-242, should be written as "If they were to get vaccinated, up to 67% (95% CI: 64-69) of unvaccinated respondents would choose Johnson & Johnson's single-dose vaccination." The sentence in lines 255-257 should be written as " The analysis revealed that reactions to the benefits of getting vaccinated were statistically substantially dependent on employment position (p=0.0142) and professional experience (p=0.0248)."  The sentences in lines 260-264 should be paraphrased as follows " They also saw it as the only option to terminate the pandemic. However, nurses were more concerned with protecting their loved ones, patients, and themselves. Unlike the doctors, they considered the idea of going to the theatre, the gym, and the cinema as well as seeing family and friends." The sentence in lines 280-282 should be written as " Almost half of those who have been vaccinated (45%, 280 95% CI: 43-47) do not communicate their opinions with patients and do not advocate or discourage them from getting vaccinated."

The sentences have been changed (line 220-291)

  • In subsection 4.4. The sentence in lines 302-303 should be paraphrased as "During the COVID-19 pandemic, 68% of the participants (95% CI: 65-70) reported having work-related anxiety regularly, and 34% (95% CI: 33-35) reported having it daily." The sentence in lines 341-343 should be corrected and paraphrased as " The ways of dealing with anxiety varied slightly based on the intensity of anxiety. People who experienced moderate or high levels of anxiety tended to cope with it by listening to music more frequently than those who had low levels of anxiety (p=0.040)."

The sentences have been changed (line 295-337)

  • In Discussion. In lines 346-348, the sentence should be written as " Infectious diseases, including SARS-CoV-2, have been and continue to be a major social problem, but more importantly, a health problem - both physically and mentally. Even if we do not become infected, we are nevertheless exposed to the psychological effects of the COVID-19 pandemic. The emergence of severe infectious diseases causes increased anxiety and fear throughout society, as seen by earlier MERS, H1N1 and SARS epidemics. As a result of the increased number of infections and deaths caused by infections, society is terrified of death, infected people and infection itself. All these circumstances led to the elevated levels of anxiety and fear." The sentence in lines 362-363 should be written as follows " After considering various reasons, 85% of respondents decided to get vaccinated against COVID-19." The sentences in lines 377-380, can be written as follows " Despite the large number of vaccinated personnel, the vast majority, up to 80%, believe that vaccination is not required to execute the duties of a doctor or nurse, and 50% consider that mandating vaccination obligations on healthcare workers violates human rights."  The sentence in line 383-383, should be paraphrased as "This allegations condradicts various statements, including those made on the Ministry of Health website, which argue that mandating vaccination among healthcare workers is the only way to protect them, citing other EU countries, such as France, Greece, Latvia and Italy." The sentence in line 390-393 should be paraphrased as "Collective responsibility appears to be one of the most important motivators for medics who desire to work safely. However, respondents justified their decision to get vaccinated with reasons such as fear of severe illness, protecting their families, or believing that this was the only way to end the COVID-19 pandemic." The sentence in lines 408-410 should be paraphrased as "Most unvaccinated medics are hesitant to get vaccinated due to the short duration of the study, as shown by research by Ledda et al." The sentence in lines 424-426 should be written as "General practitioners participating in the European multi-country study raised worries about insufficient information on vaccine safety and the possibility of overexposure." The statement in lines 478-480 does not make it obvious what it is trying to say. The sentence in lines 511-512 should be written as "To summarize, the pandemic had a significant impact not only on physical but on mental health, resulting in a variety of physical and mental burdens. The COVID-19 pandemic, which has the characteristics of a catastrophic disaster, created a difficult situation that caused fear among medical personnel."

The sentences have been changed (line 339-500)

  • In  Conclusion (section 7). The sentences in lines 530-540 should be written as "Almost all of the questioned doctors chose to get vaccinated, but nurses had a substantially lower percentage of vaccinated people, implying that work status has a considerable impact on the decision to get vaccinated against COVID-19. The profession is the most imporatnt factor in determining attitudes towards vaccinations against COVID-19. In their answers, doctors focused mostly on medical statements, stressing that their decision to get vaccinated contributed to achieving population immunity and that vaccines are the most effective method of preventing infections. They also believed that it was the only way to terminate the pandemic. The nurses, however, were more concerned with  protecting their loved ones, patients and themselves. Unlike the doctors, they considered the possibility of vacation, going to the gym, cinema, theatre, as well as meeting family and friends".  The sentences in lines 541-544 should be written as follows "The gender of the respondents differed greatly in their attitudes about immunisations. Women were far more likely than men to declare that they got vaccinated because they didn't want to endanger their families and patients; they wanted to quit wearing protective masks and return to normality." The sentence in lines 550-555 should be paraphrased as follows "The vast majority of medical workers, who based their expertise on peer-reviewed studies, opted to get vaccinated. They justified their decision primarily by increasing their chances of a milder course of the disease if they contracted it, which is consistent with the majority of the research done thus far, confirming the hypothesis that medical workers' knowledge of COVID-19 vaccinations influences their decisions to get vaccinated." The sentences in lines 556-558 should be paraphrased as "During the COVID-19 epidemic, medical workers often experienced moderate anxiety. On average, women felt more anxious than males. Younger people had slightly higher levels of anxiousness."

The sentences have been changed (line 517-545)

  • It appears to me that discussion (particularly the first half) may be condensed/shortened to make it more appealing to readers. The same should be done with the conclusions.

The discussion and conclusions have been slightly shortened

  • The error bars in the histograms in Figures 1 and 2 are required. In Tables and in several places throughout the manuscript's text, commas in numbers must be replaced by dots.

Missing figure 1 added (line 236).

Weaknesses of the study and solutions for the future have been added.

I hope that the changes made are satisfactory and this will allow publication. I am asking you to take into account the positive comments of the reviewers that this is an interesting study and a good study.

Sincerely

Authors

Reviewer 3 Report

Comments and Suggestions for Authors

For public health decision makers, the purpose of this study is relevant and could help design adequate strategies in the management of infectious disease epidemics.

To improve the proposed text of publication, i recommand the following:

- To get consistent figures on  the number of study respondents in the summary and the section 3.2 ( 385 and not 485)

- To make the introduction more focused on the study purpose, including some references from the discussion section. The introduction has to explain the knowledge gaps and to which extent , a relevant research will meet the new information expectations

- To be explicit on the location ( the country, the area covered, the health facilities involved )

- To be clear on the time frame: for exemple, were the vaccination made after or before the COVID sickness

- To decide whether to keep or drop the findings from the retest group ( very high level of no response among doctors....)

-To look after potential determinants of anxiety linked with existing co morbidities, high risk exposure, social responsibilities, mental vulnerability.;

-To develop the Section on Study strenghts and limitations ( lessons learnt from this research experience). The survey size is not the only weakness: but also the cross sectionnal nature, the lack of analysis on the working environment, the impact of Public health strategies as effectively implemented....

Author Response

Dear Reviewer,

I would like to thank for their comments. After analysing all the comments, I made the following changes:

  • To get consistent figures on  the number of study respondents in the summary and the section 3.2 ( 385 and not 485)

Editing error corrected (line 136-137)

  • To make the introduction more focused on the study purpose, including some references from the discussion section. The introduction has to explain the knowledge gaps and to which extent , a relevant research will meet the new information expectations

The introduction was slightly shortened, some content from the discussion was moved to strengthen the purpose of the research, and information about knowledge gaps was added (line 46-119)

  • To be explicit on the location ( the country, the area covered, the health facilities involved )

Information added (line 129-131)

  • To be clear on the time frame: for exemple, were the vaccination made after or before the COVID sickness

Information regarding the incidence and timing of vaccinations that were included in the study can be found in sections 4.2 and 4.3 (line 206-291)

  • To decide whether to keep or drop the findings from the retest group ( very high level of no response among doctors....)

Information was added that the results were not included in the study after repeated examination due to the small number of doctors (line 174-175)

  • To look after potential determinants of anxiety linked with existing co morbidities, high risk exposure, social responsibilities, mental vulnerability.

The study did not include obtaining such information. But we showed this as a weakness of the study and our readiness to continue research with an improved tool.

  • To develop the Section on Study strenghts and limitations ( lessons learnt from this research experience). The survey size is not the only weakness: but also the cross sectionnal nature, the lack of analysis on the working environment, the impact of Public health strategies as effectively implemented....

We added weaknesses in the study (line 502-515)

I hope that the changes made are satisfactory and this will allow publication. I am asking you to take into account the positive comments of the reviewers that this is an interesting study and a good study.

Sincerely

Authors

Reviewer 4 Report

Comments and Suggestions for Authors

Dear Editor, Dear Authors,

Thank you for the opportunity to review the manuscript entitled ‘Anxiety Levels among Healthcare Workers during the COVID-19 Pandemic and Attitudes Towards COVID-19 Vaccines’.

The submitted manuscript addresses an important topic on how healthcare workers reacted on the global threat caused by COVID-19, in my opinion, however, it suffers several issues and inconsistencies. Moreover the authors tend to overestimate their results. Therefore the manuscript does not reach scientific soundness required for publication.

The following should be considered/clarified:

*the main issue is why the authors did not use any of the COVID-19 anxiety scales typically prepared to assess anxiety associated with COVID-19, which were known at time they run their study (like e.g. CAS).

*I suggest also using a scale to measure vaccination attitudes / vaccine hesitancy to get some comparable results (see at least Oduwole, E. O., Pienaar, E. D., Mahomed, H., & Wiysonge, C. S. (2022). Overview of Tools and Measures Investigating Vaccine Hesitancy in a Ten Year Period: A Scoping Review. Vaccines, 10(8), 1198. https://doi.org/10.3390/vaccines10081198).

*The STROBE guidelines are for reporting. There are not guidelines how to lead or to conduct1 the study.

*The convenience sampling is the sampling method which does not guarantee representativeness. Moreover, convenience sampling should not be used, as it is non-random sampling, which leads to sampling bias.

*the inclusion criteria are the criteria which characterize the target population, and the fact whether someone sign or not a consent has nothing to that characteristic (it is hard to conclude that study results refer only to those health care workers which would sign a consent to participate in a study).

*the exclusion criteria are not criteria which are opposite to inclusion criteria. The study flow is … check inclusion criteria, if ok, than verify whether the patient has any exclusion criteria.

*I do not understand why depression was not considered either as a exclusion criterion or an important covariate (as up to 30% healthcare workers meet likely diagnostic criteria for clinical depression … see the AMADEUS survey)

*why medications for anxiety were not considered … the access to drugs among healthcare workers is clearly easier than in general population, and it seems to be quite reasonable to take drugs for some health problems experienced

*the calculation of the required sample size is very confusing. The procedure requires decision on point estimate(s) and the level of precision and then calculate considering the level of confidence. Provide, please, the reference and rationale for 1 item for 10 study participants

*there are two different numbers of respondents in the manuscript, so what was the true number?

*authors state ‘the research has an acceptable level of internal consistency and test-retest reliability, and well as construct validity’. Firstly, these features are for the research tool as a scale but not for research itself. Secondly, provide please, the results for consistancy, reliability and validity.

*authors decided to use the chi-squared test but they did not verify (no information about) whether the assumptions of the test have been met or not.

*the sentence “The standardized level of correlation α=0.05 allowed for the development of statistical inference” has no sense. The correlation is quite different type of an analysis. Probably, authors decided to use the type-I error of 0.05% (which reflects the ‘cut-off’ for statistical significance).

*the concept of statistical significance is Yes/No (binary), meaning the result is statistically significant or not. There is no sense to name / categorize results as ‘a statistically significant’, ‘a highly significant’, or ‘a very highly significant’

*the term ‘relationship’ refers causality … however (as also mentioned by authors) the study design does not mandate to state on causality. This should be corrected in the manuscript.

*tab.1 What is ‘seniority’ and ‘do 10 years’?

*tab.2. The table probably does not present results by Student’s t-test as mentioned in the table … the same comment is for tab.4

*the 95% CI for proportions presented on the page 9 seem to be calculated improperly. Authors state “More than half of the respondents (52%, 95% CI: 50-52) admitted that” … firstly, the 95% CI is around the point estimate therefore the upper bound of 52 is improper, next  … 52% out of 385 gives 200, and 95% CI for the sample of 385, 200 participants representing the feature, 95% CI leads to the interval: 46.83%-57.04% by binomial exact calculation (or to 46.96%-56.94% by normal approximation to the binomial calculation”. Other provided results suffer the same issue.

*page 16 – line 523: the sample is representative or not, it is hard to understand what the authors meant by “more representative”

*authors state “the number of participants included in the study was adequate to detect even relatively small effects” but this was not shown, and look like overestimation.

*conclusion 1: job positions as a factor associated with getting vaccine were not compared in this study therefore it is hard to conclude on its role on vaccination

*conclusion 4: “Medical staff most often experienced moderate anxiety during the COVID-19 pandemic” … but referring to Tab.9 the most often is ‘low level’, so what results the authors refer to?

I suggest correcting carefully all the inconsistencies and questionable results.

Regards,

Reviewer

Comments on the Quality of English Language

I suggest reviewing the paper by a native speaker to correct some grammar errors and improve a fluency. Some sentences look like a direct translation from an original language

Author Response

Dear Reviewer,

I would like to thank for their comments. After analysing all the comments, I made the following changes:

  • the main issue is why the authors did not use any of the COVID-19 anxiety scales typically prepared to assess anxiety associated with COVID-19, which were known at time they run their study (like e.g. CAS).

The Trait Anxiety Scale SL-C used in the study. We showed this as a weakness of the study. We want to continue the research by adding validated tools.

  • I suggest also using a scale to measure vaccination attitudes / vaccine hesitancy to get some comparable results (see at least Oduwole, E. O., Pienaar, E. D., Mahomed, H., & Wiysonge, C. S. (2022). Overview of Tools and Measures Investigating Vaccine Hesitancy in a Ten Year Period: A Scoping Review. Vaccines, 10(8), 1198. https://doi.org/10.3390/vaccines10081198).

We showed this as a weakness of the study. We want to continue the research by adding validated tools.

  • The STROBE guidelines are for reporting. There are not guidelines how to lead or to conduct1 the study.

These guidelines helped us organize the research and create the research report

  • The convenience sampling is the sampling method which does not guarantee representativeness. Moreover, convenience sampling should not be used, as it is non-random sampling, which leads to sampling bias.

There was a translation error, we have corrected the method (line 129-130)

  • the inclusion criteria are the criteria which characterize the target population, and the fact whether someone sign or not a consent has nothing to that characteristic (it is hard to conclude that study results refer only to those health care workers which would sign a consent to participate in a study).

The entry has been clarified (line 129-137)

  • the exclusion criteria are not criteria which are opposite to inclusion criteria. The study flow is … check inclusion criteria, if ok, than verify whether the patient has any exclusion criteria.

The entry has been clarified (line 129-137)

  • I do not understand why depression was not considered either as a exclusion criterion or an important covariate (as up to 30% healthcare workers meet likely diagnostic criteria for clinical depression … see the AMADEUS survey)

We did not expect such results, given the knowledge we had. In continuation of the study, we took into account all the Reviewer's suggestions.

  • why medications for anxiety were not considered … the access to drugs among healthcare workers is clearly easier than in general population, and it seems to be quite reasonable to take drugs for some health problems experienced

We did not ask questions about the medications used. In continuation of the study, we took into account all the Reviewer's suggestions.

  • the calculation of the required sample size is very confusing. The procedure requires decision on point estimate(s) and the level of precision and then calculate considering the level of confidence. Provide, please, the reference and rationale for 1 item for 10 study participants

Recording corrected (section data and method)

  • there are two different numbers of respondents in the manuscript, so what was the true number?

Recording corrected (line 135-137)

  • authors state ‘the research has an acceptable level of internal consistency and test-retest reliability, and well as construct validity’. Firstly, these features are for the research tool as a scale but not for research itself. Secondly, provide please, the results for consistancy, reliability and validity.

Recording corrected (section data and method)

  • authors decided to use the chi-squared test but they did not verify (no information about) whether the assumptions of the test have been met or not.

The entry in the statistical analysis was clarified (line 187-197)

  • the sentence “The standardized level of correlation α=0.05 allowed for the development of statistical inference” has no sense. The correlation is quite different type of an analysis. Probably, authors decided to use the type-I error of 0.05% (which reflects the ‘cut-off’ for statistical significance).

The entry in the statistical analysis was clarified (line 187-197)

  • the concept of statistical significance is Yes/No (binary), meaning the result is statistically significant or not. There is no sense to name / categorize results as ‘a statistically significant’, ‘a highly significant’, or ‘a very highly significant’

The entry in the statistical analysis was clarified (line 187-197)

  • the term ‘relationship’ refers causality … however (as also mentioned by authors) the study design does not mandate to state on causality. This should be corrected in the manuscript.

The entry in the statistical analysis was clarified (line 187-197)

  • 1 What is ‘seniority’ and ‘do 10 years’?

A language error has been corrected (table 1)

  • 2. The table probably does not present results by Student’s t-test as mentioned in the table … the same comment is for tab.4

Editorial errors corrected (results section)

  • the 95% CI for proportions presented on the page 9 seem to be calculated improperly. Authors state “More than half of the respondents (52%, 95% CI: 50-52) admitted that” … firstly, the 95% CI is around the point estimate therefore the upper bound of 52 is improper, next  … 52% out of 385 gives 200, and 95% CI for the sample of 385, 200 participants representing the feature, 95% CI leads to the interval: 46.83%-57.04% by binomial exact calculation (or to 46.96%-56.94% by normal approximation to the binomial calculation”. Other provided results suffer the same issue.

Editorial errors corrected (results section)

  • page 16 – line 523: the sample is representative or not, it is hard to understand what the authors meant by “more representative”

this entry has been removed (line 501-515)

  • authors state “the number of participants included in the study was adequate to detect even relatively small effects” but this was not shown, and look like overestimation.

this entry has been removed (line 501-515)

  • conclusion 1: job positions as a factor associated with getting vaccine were not compared in this study therefore it is hard to conclude on its role on vaccination

Completed the information (line 218-219)

  • conclusion 4: “Medical staff most often experienced moderate anxiety during the COVID-19 pandemic” … but referring to Tab.9 the most often is ‘low level’, so what results the authors refer to?

In self-assessment, the respondents experienced a moderate level of anxiety. According to the scale, low. The entry has been corrected (abstract, line 542-544)

  • I suggest reviewing the paper by a native speaker to correct some grammar errors and improve a fluency. Some sentences look like a direct translation from an original language

The manuscript was revised by a native speaker

I hope that the changes made are satisfactory and this will allow publication. I am asking you to take into account the positive comments of the reviewers that this is an interesting study and a good study.

Sincerely

Authors

Round 2

Reviewer 1 Report

Comments and Suggestions for Authors

 None to add

Author Response

Dear Reviewer,

We would like to thank you very much for your favorable review.

Sincerely

Authors

Reviewer 2 Report

Comments and Suggestions for Authors

After reading the revised version of the manuscript, it is evident that authors made corrections in virtually all the tex places I have suggested, and the text, in my opinion, has now become substantially more appealing. I recommend accepting this article for publication after some minor corrections (a small edits are required for the numbers in Tables 3, 4, 5, and 6).  

Author Response

Dear Reviewer,

We would like to thank you very much for your favorable review. We have made corrections in tables 3, 4, 5, 6.

Sincerely

Authors

Reviewer 3 Report

Comments and Suggestions for Authors

Thank you for your reply and improvements .

Your amendments and additional information are relevant.

The section on Limitations and Strengths is still quite succinct.

Author Response

(The authors gave the same response as above.)

Reviewer 4 Report

Comments and Suggestions for Authors

Dear Authors,

Dear Editor,

After the revision, the clarity of the study results described in the manuscript has been significantly improved. In the current form, in my opinion, the manuscript may be considered for publication.

Reviewer

Author Response

(The authors gave the same response as above.)
